# Benign Bone Tumors: An Overview of What We Know Today

**DOI:** 10.3390/jcm11030699

**Published:** 2022-01-28

**Authors:** Sara De Salvo, Vito Pavone, Sebastiano Coco, Eleonora Dell’Agli, Chiara Blatti, Gianluca Testa

**Affiliations:** Department of General Surgery and Medical Surgical Specialties, Section of Orthopaedics, A.O.U. Policlinico Rodolico-San Marco, University of Catania, 95123 Catania, Italy; sarads94@hotmail.it (S.D.S.); vitopavone@hotmail.com (V.P.); sebastianococo1989@gmail.com (S.C.); dellagli.eleonora@hotmail.it (E.D.); chiara.blatti@gmail.com (C.B.)

**Keywords:** benign bone tumors, nonmalignant bone tumors, bone-forming tumors, cartilage-forming tumors, bone cysts, fibrous bone tumors

## Abstract

Nonmalignant bone tumors represent a wide variety of different entities but maintain many common features. They usually affect young patients, and most can be diagnosed through imaging exams. Often asymptomatic, they can be discovered incidentally. Due to their similarities, these tumors may be challenging to diagnose and differentiate between each other, thus the need for a complete and clear description of their main characteristics. The aim of this review is to give a picture of the benign bone tumors that clinicians can encounter more frequently in their everyday work.

## 1. Introduction

Approximately 40% of the skeletal and musculoskeletal proliferative lesions do not determine metastasis and are related to a good survival rate [1]. These groups of proliferative lesions are generally defined as benign bone tumors (BBTs). However, they can occur in any part of the skeleton and can still be dangerous as they may grow and compress healthy tissues. 

This definition includes a wide variety of diseases, which vary in terms of incidence, clinical presentation, and possess a diverse array of therapeutic options. Many classifications have been proposed. The last WHO classification considers the biological behavior of the lesions, dividing benign bone tumors into “intermediate (locally aggressive)” and “intermediate (very rarely metastatic)”. Other classifications are based on the type of matrix production or otherwise histological, clinical, and radiological criteria.

The most frequent BBTs are osteochondroma, osteoma, osteoid osteoma, osteoblastoma, giant cell tumor, aneurysmal bone cyst, fibrous dysplasia, and enchondroma. These tumors are further divided into categories based on their cell type: bone-forming, cartilage-forming, as well as connective tissue and vascular [2,3]. The incidence of BBTs is hard to calculate since they are often asymptomatic and difficult to detect [2]. Some forms of BBTs are sporadic, and due to their low incidence, they will not be discussed.

Overall, they arise in people less than 30 years old, often triggered by the hormones that stimulate growth (like osteochondroma).

This review aims to discuss different types of common BBTs, based on the best-published literature available about the topic.

## 2. Bone-Forming Tumors

### 2.1. Osteoid Osteoma

#### 2.1.1. Introduction

A bone-forming tumor is a benign osteoblastic tumor, accounting for 12% of benign tumors of the bone [4]. It can develop at any age, even before 3 years old [5,6], but its typical presentation is in the second decade, more frequently in boys (two to four times more than girls) [7]. Usually solitary, there have been reports about rare metachronous presentations [8]. It is composed of a nidus surrounded by sclerotic tissue whose dimensions are generally less than 1.5 cm in diameter. If the lesion has a longer diameter, it can be considered an osteoblastoma, similar in genetics and morphology [9]. The debate between the differentiation of these two separate entities is still going on in the scientific community.

#### 2.1.2. Sites of Development

Bone-forming tumors mainly affect lower limb long bones, but a frequent site of presentation is the spine, especially the lumbar district [10].

#### 2.1.3. Micro-Macroscopic Features

The histopathological exam shows a central nidus composed of tiny osteoid islands lined by osteoblasts. Pain is attributed to unmyelinated nerve fibers present within the nidus [11]. The area peripheral to the nidus appears clearer because of osteoclastic resorption, and dense sclerotic tissue surrounds the nidus. According to some studies, the nidus produces prostaglandins, and it may also produce osteocalcin [12,13]. Its genetic features are similar to those of osteoblastoma; in fact, osteoid osteoma expresses Runx2 and Osterix, transcription factors involved in osteoblastic differentiation [14].

#### 2.1.4. Clinical Features

Bone-forming tumors present with increasing pain that may worsen at night. One of their main characteristics is that this pain is resolved with NSAIDs (prostaglandin inhibitors) in less than half an hour: if this does not occur, it is very likely not to be an osteoid osteoma and other options have to be investigated [15]. Moreover, bone-forming tumors can present with swelling, erythema, tenderness, and muscular atrophy [16]. The clinical symptomatology may differ based on the presentation site. In patients affected by a spine osteoid osteoma, this can result in scoliotic posture; therefore, differential diagnosis is fundamental [17]. This lesion may affect open physis, causing lengthening or angular deformity of the affected bone. Furthermore, if it affects the hip joint, causing hip impingement; therefore, an accurate evaluation of the femoroacetabular impingement becomes mandatory. When developing intra-articular or juxta-articular, the lesion can cause synovitis [18].

#### 2.1.5. Imaging

In X-ray examination, a bone-forming tumor appears as a nidus surrounded by sclerotic tissue. It may be hard to identify when it is intra-articular, therefore requiring further look examination. The second-level exam is a CT scan, which displays a small, defined nidus surrounded by a sclerotic reaction which may contain calcifications. Bone scintigraphy usually shows arterial phase uptake in the nidus due to its high vascular concentration and lower concentration in the reactive surrounding bone. There is a typical sign called the double-density sign which is diagnostic. It is fundamental to pay attention to growth plates since they can obscure the signal if the osteoid osteoma is in their proximity; therefore, it can be helpful to evaluate the contralateral side. Other imaging techniques may help the diagnostic workup, but those must be considered case by case [19,20,21] (Figure 1).

#### 2.1.6. Differential Diagnosis

For differential diagnosis, it is essential to exclude osteoblastoma that presents with no pain at all or dull pain, responding less to NSAIDs, and it is usually more extensive than the osteoid osteoma and with a more complex morphology. Osteoid osteoma presentation can resemble that of subacute or chronic osteomyelitis, but those typically extend from growth plates.

#### 2.1.7. Treatment and Prognosis

The natural history of the tumor is of spontaneous resolution within 6 to 15 years, and, if not symptomatic, watchful waiting is the suggested approach. On the other hand, NSAIDs are the drug of choice if symptoms are present. Studies have reported that treatment with NSAIDs or aspirin could reduce the time of healing by 2 or 3 years. In case of symptomatic lesions, such as those causing scoliosis, length discrepancy, and other morpho-functional disorders, those can be approached with surgical resection, radiofrequency, cryotherapy, and MRI-guided high-intensity focused ultrasound. It is not necessary to excise the sclerotic margins; the focus of the resection must be the removal of the nidus. Recurrences can happen if, after resection, the nidus is not entirely excised [15,22,23,24].

### 2.2. Osteoblastoma

#### 2.2.1. Introduction

Accounting for approximately 14% of benign bone tumors, osteoblastoma is a bone-forming neoplasm that affects primarily young adults (mean age 20 years), predominantly males (two to three times more than females). The tumor is more aggressive as the age of presentation is [25]. Like osteoid osteomas, they are composed of a central nidus surrounded by sclerotic tissue. Although the nidus is more vascularized and the lesions less organized, it does not produce prostaglandins, and it carries a lower amount of nerve fibers. The average size is around 3 cm, but it can grow to 15 cm of diameter [26]. 

#### 2.2.2. Sites of Development

Osteoblastoma most frequently arises from the posterior elements of the spine and the sacrum. Long tubular bones of the lower limb are another frequent site of presentation with the mandible, and in this last location, it is referred to as cementoblastoma [7,27].

#### 2.2.3. Micro-Macroscopic Features

An osteoblastoma is a rich vascularized lesion in red/brown color. It is usually sharply distinguished from the bone marrow by a border, and the margins on the cortical surface push against it. Usually, they are separated from other bone structures by a fibrous tissue layer. The borders permeate more as the tumor becomes more aggressive. An essential feature of this lesion and OO is the formation of immature and osteoid bone trabeculae [28]. It is possible to differentiate osteoblastoma from osteosarcoma immunohistochemically because the former presents with a beta-catenin nuclear staining, while the latter presents with a cytoplasmic or membranous one [28]. As mentioned for osteoid osteoma, osteoblastoma is characterized by genetic mutations of FOS and FOSB [29].

#### 2.2.4. Clinical Features

As with most benign tumors, osteoblastoma tends to grow slowly and be asymptomatic for most patients: this clinical feature may help the differential diagnosis with osteoid osteoma. When present, symptoms may include dull, localized pain, with soft tissue swelling. As in other cases, symptoms may vary based on the presentation site. It can cause nerve root compression, postural scoliosis, and muscle weakness when affecting the spine. Patients with lower extremity affection may present with limp [30,31]. Toxic osteoblastoma is a rare variant of this neoplasm with systemic symptoms, such as fever, anorexia, and diffuse periostitis [32].

#### 2.2.5. Imaging

The evaluation algorithm of osteoblastoma includes plain radiographs, CT scans, and MRI. Its appearance in plain radiographs can have different patterns. It can be similar to osteoid osteoma but greater (>2 cm) and surrounded by less reactive sclerosis. Benign periosteal reaction is a standard feature. Lesions developing in the axial skeleton can mimic ABCs. Furthermore, mainly aggressive osteoblastomas can resemble malignant lesions when more extensive in size (>4 cm), causing disruption and cortical thinning.

#### 2.2.6. Differential Diagnosis

The differential diagnosis includes many pathologies, such as osteoid osteoma, osteomyelitis, stress fracture, osteosarcoma, giant cell tumor of bone, and aneurysmal bone cyst. To correctly identify the disease, advanced radiographic imaging might be necessary [33,34]. 

#### 2.2.7. Treatment and Prognosis

Differently from osteoid osteoma, osteoblastoma may have an aggressive development: for this reason, it is preferably treated surgically. The procedure of choice varies based on the aggressiveness grade of the lesion. The most frequent options are curettage and resection en bloc, depending on the tumor’s aggressiveness. Intralesional curettage has a higher rate of recurrences, but it is less invasive, so it is used if the suspicion of malignancy or evolution toward it is low. On the other hand, en bloc resection leads to a lower rate of recurrences than other techniques; therefore, it is used in more aggressive lesions. Other therapeutic tools, such as radiotherapy or chemotherapy, have not proven effective for this tumor. It is fundamental to continue the patient follow-up after surgery with imaging exams, to exclude eventual recurrences. If untreated and enlarging, an osteoblastoma can damage the adjacent bone or nervous structures [35,36,37].

### 2.3. Giant Cell Tumor

#### 2.3.1. Introduction

Giant cell tumors represent 15–20% of benign bone neoplasms in the United States, having even higher incidence rates in other countries, such as China [38]. Unlike other benign tumors, they can result metastases (pulmonary in most cases) and are locally aggressive, causing osteolysis. It affects primarily young adults around the third decade of life [39,40].

#### 2.3.2. Sites of Development

Giant cell tumors frequently develop from the meta-epiphyses of the long bones, especially all four bones that make up the knee joint, particularly the proximal tibia and distal femur [41,42]. 

#### 2.3.3. Micro-Macroscopic Features

In the histopathological exam, GCTB is friable and of red/brown color. The giant cells are multinucleated and immersed in a stromal background of monocytes and spindle cells. The pattern can be variable, including hemosiderin, necrosis, and hemorrhage. The neoplastic cells are the mononuclear spindle cells. [43] Studies on its genetics have shown that 20% of GCTB have p53 overexpressed, and those are associated with neoplasm recurrence and metastases [44]. Other findings have been increased telomerase activity and their shortening prevention [45]. A study found that 54% of GCTB had a 20q11 amplification [46]. RANKL overexpression is the molecular feature most represented in GCTB, particularly in osteoblast-like mononuclear stromal cells, recruiting osteoclast cells. The latter would then cause osteolysis. Histone modification, found in over 90% of these tumors, may be one of the starting points of neoplastic formation [47,48].

#### 2.3.4. Clinical Features

Patients typically present with pain. At physical examination, it can be possible to find tenderness to palpation, swelling, and, in case of joint proximity, joint effusion. Antalgic gait may also be a frequent finding. Pathologic fractures can be an indirect sign of the presence of the lesion, often involving the joints [49].

#### 2.3.5. Imaging

The imaging workup comprehends plain X-rays, CT scan, and MRI exams. Simple X-rays can show various patterns, typically a radiolucent meta-epiphyseal lesion. One of the most used classifications was created by Campanacci in 1987 (Table 1).

This classification was made to influence the extent of surgery. Even if it simplifies the diagnostic workup, other histopathological and genetic factors must be evaluated to clarify the tumor’s aggressiveness. CT can be used to define the lesion and its borders further. MRI findings are often nonspecific, with the T1-weighted sequence showing a decrease in signal intensity and increase in the T2-weighted signal: these sequences can also spot hemosiderin deposition, a frequent finding in GCTB, indicated by low signal intensity. Another nonspecific but frequent feature of GCTB in scintigraphy is the “donut sign” due to the intense uptake of radionuclide in the periphery and the necrosis in the center of the lesion. Chest CT imaging is suggested in recurrent GCTB to spot eventual pulmonary metastases [50,51].

#### 2.3.6. Differential Diagnosis

Differential diagnosis includes benign and malignant tumors rich in giant cells and osteoclasts, such as brown tumors associated with hyperparathyroidism and metastases of other body tumors [52].

#### 2.3.7. Treatment

The gold standard for GCTB is surgery. A frequent approach consists of intralesional curettage and cavity filling with PMMA (polymethylmethacrylate). Extensive bone resection is typically performed if the tumor is located in expendable bones such as fibula or distal ulna. In the case of extraosseous extension and metastases, different approaches, such as denosumab, radiotherapy, bisphosphonates, and chemotherapy, can be evaluated. There is no consensus about the follow-up, but these patients must be periodically checked on with imaging exams [53,54,55,56]. Metastases affect 2–3% of GCTB and develop almost exclusively from the lungs (especially if the primary tumor arises from the spine). These metastases though, maintain the histological and morphological pattern of the tumor of origin, and therefore are often referred to as “benign” pulmonary implants. These are most frequently found in recurrent lesions [57,58].

## 3. Cartilage-Forming Tumors

### 3.1. Osteochondroma

#### 3.1.1. Introduction

Osteochondroma represents 20–50% of benign tumors [59], being the most common bone benign tumor, and 10–15% of all tumors in general. It typically develops in growing bones. For this reason, according to some studies, they probably arise from the epiphyseal growth plate. They affect males more often than females. It appears as a bony projection covered by cartilage on the external surface of the bone. Single or multiple lesions can occur in multiple hereditary osteochondromas [60]. The single osteochondroma (85%) usually presents around 20 years old. It is hypothesized that the cause of this tumor can be the herniation of a fragment of the growth plate through the periosteum, which forms a sessile or pedunculated lesion. This herniation can occur spontaneously or after trauma or radiation exposure. However recent studies have found tumor suppressor exostine-1 (EXT1) mutation [61]. This lesion could be found incidentally for other exams. In the multiple osteochondromas, the age of onset is earlier, and typically, symptoms are more evident. Most of these lesions are caused by autosomal dominant inheritance with incomplete penetrance with an incidence of 1/50,000–100,000 [61]. The mutation of tumor suppressor genes exostin-1 and exostin-2 is frequent. In both cases, the lesions could increase their dimensions during maturing phases; a malignant lesion should be suspected when the skeletal maturity is complete and the lesion increases. 

#### 3.1.2. Sites of Development

The mainly involved bone segment is the metaphysis. The distal femur is the most common location, followed by proximal tibia and humerus. Osteochondromas can involve the spine. 

#### 3.1.3. Micro-Macroscopic Characteristics

The cartilage that covers the bone is bluish and typically has dimensions of 1–3 cm in children and a thickness of few millimeters in adults (if the cartilage thickness is >1 cm in adults, malignancy must be suspected) [62]. A pathognomonic feature of osteochondroma is lesion continuity from the cortex to the medulla [59].

#### 3.1.4. Clinical Features

A single osteochondroma presents a painless mass, whereas multiple osteochondromas may cause multiple bone deformities, bowing of the extremities, short limbs, short stature, leg length discrepancy, coxa valga, or valgus knee [59].

#### 3.1.5. Imaging 

At X-ray examination, osteochondroma is usually localized in the metaphysis, and its cortex is in continuity through a marrow cavity with the originating bone cortex. In the case of symptomatic patients, further investigations could be necessary with MRI, CT, or US. MRI is helpful to evaluate the cartilage cap and bone thickness in adults and the possible complications, such as bursae, neurological, and vascular involvement [59]. Scintigraphy shows enhancement in benign and malignant lesions; for this reason, it is not a valuable test for diagnostic purposes (Figure 2).

#### 3.1.6. Differential Diagnosis

This disease must be differentiated from subungual exostosis (Dupuytren’s exostosis), in which lesions develop on the back of the phalanx, close to the nail and can be painful and ulcerated, in addition to hemimelic dysplasia of the epiphysis (Trevo disease), presence of multiple osteochondromas in the lower limbs, and parosteal osteosarcoma, a subtype of osteosarcoma arising from the surface of long bones. Radiographs show a large, lobulated, and dense bone mass; in the advanced stages, the lesion can infiltrate the medullary space. In all these cases, there is no lesion continuity from the cortex to the medulla.

#### 3.1.7. Treatment and Prognosis

Asymptomatic single osteochondromas follow over time [63]. In case of symptomatic single osteochondromas or concern for malignant transformation, excision is indicated. The recurrence after incomplete excision of the cartilage cap is 2%, and it could occur only during the stages of development. Osteochondroma has a good prognosis and only 1% turns malignant; hereditary osteochondroma has a 2–5% risk of malignant transformation.

### 3.2. Enchondroma

#### 3.2.1. Introduction

Enchondroma is a nonmalignant tumor that develops inside the medullary bone cavity of endochondral origin. It is composed of hyaline cartilage. Enchondroma is probably the second most common nonmalignant primary bone tumor, after osteochondroma, accounting for 3–10% of all bone tumors and 13% of all nonmalignant bone tumors [64]. It typically presents between 15–35 years, and it occurs with the same frequency in males and females. In 40% of cases, enchondroma develops after a sporadic mutation of isocitrate dehydrogenase-1 (IDH1) and isocitrate dehydrogenase-2 (IDH2). Isocitrate dehydrogenase is an enzyme that converts isocitrate into alpha-ketoglutarate in the tricarboxylic acid cycle. This mutation induces high levels of the oncometabolite D-2-hydroxyglutarate (D-2-HG) [64]. The oncometabolite inhibits the differentiation of staminal cells during skeletal formation and stimulates the formation of chondroid tumors [65]. Mutation of parathyroid receptor-1 (PTHR1) and genetic abnormalities of chromosomes 5, 6, 7, 12, and 17 [66] are rare. Enchondromas are nonmalignant lesions, even if they can turn into malignant lesions, such as chondrosarcoma.

#### 3.2.2. Sites of Development

Ninety percent of enchondromas are solitary lesions that grow from an ectopic cartilage nest in the medullary cavity. It is assumed that enchondroma originates from remnants of growth plates trapped in bone metaphysis during growth. The trapped chondrocytes keep their chondroid properties and do not turn to bone [67]. Enchondroma develops in the metaphyseal segment of the tubular bone of the hand and foot and long bones, such as the femur and humerus. The typical site of development is the proximal metaphysis of the proximal phalanx on the ulnar side [68]. The proximal phalanges are most frequently affected, followed by the middle phalanges, metacarpals, and distal phalanges.

#### 3.2.3. Micro-Macroscopic Characteristics

Enchondroma is a nodular lesion, bluish-gray in color, translucent, and hypocellular, with a considerable quantity of hyaline cartilage without vascular components. Nuclei are regular and defined with poor mitotic proliferation [69]. Chondroid calcified matrix and necrosis of chondrocytes suggest enchondroma in adults. Enchondroma of the hand is characterized by hypercellular and cytologic abnormalities, properties that chondrosarcoma also has [70]. 

#### 3.2.4. Clinical Features

Enchondromas are generally asymptomatic, and they could be found incidentally during other exams [71]. Pathologic fracture or malignant transformation can cause pain.

#### 3.2.5. Imaging 

Radiographic features of enchondroma include solitary oval radiolucent lesion in the central area of metaphysis of the bone. In adulthood, in the middle of this lesion is possible to see calcifications with higher signal radiodensity [12]. CT is useful for detect the classic chondroid mineralization of the “arcs and rings” pattern [72]. MRI gives information on aggressive and erosive features of the lesion. Enchondroma and chondrosarcoma share the same low-intensity signal on T1 and high-intensity signal on T2. Perilesional edema is usually present around the enchondroma and absent around chondrosarcoma, simplifying the differential diagnosis [73]. Biopsy is not recommended unless it is an atypical lesion or for doubtful cases.

#### 3.2.6. Differential Diagnosis

Differential diagnosis has to be done with bone infarction, low-grade chondrosarcoma: progressive destruction of the chondroid matrix, an expanding lesion that could invade soft tissue, and pain is the predominant symptom, and tuberculous dactylitis (spina ventosa). It remains a challenge to differentiate enchondroma from chondrosarcoma. Chondrosarcoma tends to present in the axial skeleton of middle-aged adults (40–50 years old) and is often associated with insidious, progressive, and night pain. Radio-graphically it looks like a lobular lytic lesion with mineralized matrix and poorly defined margins. Enchondroma tends to present in the appendicular skeleton of young adults (30–40 years old). It is often an incidental finding and asymptomatic lesion. Radiographically it looks like a well-demarcated central lesion with the typical pattern of “arcs and rings calcification”. Enchondroma and chondrosarcoma share similar histological aspects and mutations like IDH1 and IDH2. The presence of perilesional edema in MR imaging, periostin, and alpha-methyl-acyl-CoA racemase (AMACR) in the stroma at immunohistochemistry exam allows making a possible diagnosis of chondrosarcoma [74].

#### 3.2.7. Treatment and Prognosis

Asymptomatic and small enchondroma do not require treatment, but only observation. Early or delayed surgical treatment in these cases have similar results [75]. Symptomatic Enchondroma needs simple curettage and bone grafting [76]. Bone grafting can be allogeneic, autogenic, or composed by synthetic bone substitutes. The risk of fracture increases when the lesion has a diameter >25 mm or the lesion involves >50% of the diameter of the cortex. Early surgical treatment in these cases is controversial. In case of fractures, surgery is recommended. Enchondroma has a good prognosis; it is a nonmalignant and self-limiting lesion. Recurrence after surgical treatment with curettage and bone graft is rare [77].

#### 3.2.8. Enchondromatosis

The presence of multiple enchondromas with a unilateral predominance is defined as enchondromatosis. The mutations of IDH1 and IDH2 are the most frequent [32]. This condition is typical of two diseases: Ollier disease and Maffucci syndrome. 

Ollier disease is a sporadic rare condition with a prevalence of 1 in 100,000. Enchondromas usually develop before solitary enchondroma, at around 10 years of age. Multiple enchondromas grow in the epiphyses or between the metaphysis and the diaphysis [78]. The site of development could prevent proper bone growth, enlargement of the metaphysis, and bowing of the bone. 

Maffucci syndrome is characterized by multiple enchondromas, soft tissue hemangiomas, and vascular abnormalities [79,80]. Patients affected by Maffucci syndrome often develop malignant non-skeletal lesions, such as gliomas or ovarian or pancreatic carcinomas.

In both diseases, there is a higher risk of malignant transformation in chondrosarcoma (40%) [81]; in detail, the risk is 35% higher in Ollier disease and 50% higher in Maffucci syndrome. Histological features of both conditions include hypercellularity and cellular atypia (Figure 3).

### 3.3. Chondroblastoma

#### 3.3.1. Introduction

Chondroblastoma is a nonmalignant tumor that originates from chondroblasts, the most immature cells of the cartilage. It represents <1% of all bone tumors and almost 5% of nonmalignant bone tumors [82]. It typically presents during teenage years, and it is more common in boys than girls, with a 2:1 ratio. The etiology is unknown, but it is hypothesized to be a mix of environmental and predisposing factors. The rearrangement of 5 and 8 chromosomes is common. Specific mutations (K36M) of gene H3F3B or H3F3A have recently been discovered, both codifying to H3.3 histone [83].

#### 3.3.2. Site of Development

Chondroblastoma is typically a solitary lesion that grows in teenagers with skeletal immaturity on the epiphyses or in the apophysis of long bones [84]. The proximal humerus is commonly involved, followed by distal femur and proximal tibia. 

#### 3.3.3. Micro-Macroscopic Characteristics

Chondroblastomas present pinkish-brown lesions, with occasional bleeding and cystic formations. Histologically chondroblastomas are characterized by a layer of polygonal cells with well-defined cytoplasm and round nucleus. They always present giant cells, the osteoclasts. A pericellular calcification net, defined as “chicken wire”, is pathognomonic [85]. At immunohistochemistry, chondroblastoma is positive for vimentin, neuron-specific enolase, protein S-100, SOX9, DOG1, and keratins 8, 18, and 19 [86,87].

#### 3.3.4. Clinical Features

Symptoms of chondroblastoma include constant low-grade joint pain, swelling, joint stiffness, and limp. Physical examination often shows joint effusion, decreased range of motion, and muscle atrophy [84].

#### 3.3.5. Imaging 

On radiography, chondroblastoma appears as a small and eccentric lesion, well-defined with sclerotic borders in the epiphyses. The diameter of the lesion is 3–6 cm. CT is used for highlighting the possible erosion of the cortex, calcifications of the matrix and extension to soft tissues. MRI shows the typical perilesional edema, and in post-contrast imaging, gadolinium is localized in the center of the lesion [87]. 

#### 3.3.6. Differential Diagnosis

Differential diagnosis must be made, especially with giant cell tumor, chondromyxoid fibroma, aneurysmal bone cyst, and osteomyelitis.

#### 3.3.7. Treatment and Prognosis

The treatment goal is to decrease pain, to avoid the spread of the lesion in the joint and close soft tissues, and to decrease recurrence; in order to achieve these, surgery is performed. The type of surgery depends on the bone/joint involvement, the location, and lesion stage (stage 1—latent, stage 2—active, stage 3—aggressive). The surgical options consist of complete curettage with or without bone graft, en bloc bone resection, and sometimes amputation. In addition to surgical therapy, chemical cauterization with phenol or cryotherapy is preferable to reduce the risk of recurrence. After surgery, follow-up is essential. In case of recurrence en bloc bone resection is the best choice [88]. Radiofrequency could be an alternative treatment if surgery cannot be performed. Adjuvant chemotherapy and radiotherapy do not have a well-documented role in the treatment of chondroblastomas [84]. Chondroblastoma has a good prognosis after surgical treatment. Recurrence has a rate of 8.3–21%, and it increases if the surgery is inadequate or if the lesion is localized in the hip or pelvis [89,90].

## 4. Fibrous Lesions

### 4.1. Fibrous Dysplasia

#### 4.1.1. Introduction

FD is a benign bone lesion characterized by proliferative fibrous tissue that replaces the normal bone matrix. It represents approximately 5–7% of benign bone tumors [91]. Its prevalence is similar in both sexes. There are two types of FD: monostotic fibrous dysplasia (MFD) or polyostotic fibrous dysplasia (PFD). MFD accounts for 70%. When polyostotic, lesions may be unilateral or bilateral. FD is associated with endocrine disorders (MAS) in only 2–3% of cases. 

#### 4.1.2. Sites of Development 

Lesions most often present in the long bones of the legs, arms, ribs, and pelvis, as well as in craniofacial bones. The spine is involved in 1.4–5.5% of FD lesions. MFDs generally grow up to complete skeletal development and then cease, on the contrary, PFDs continue to expand also after full skeletal maturity is established, creating weak areas that may result in pathological fractures [92]. PFD in 2–3% of cases may be associated to endocrine disorders, such as McCune-Albright syndrome (MAS) [93] or Mazabraud syndrome, a rare benign condition characterized by the association of single or multiple intramuscular myxomas with polyostic fibrous dysplasia. FD can also be associated with the development of hypophosphatemic osteomalacia due to the increased hormone regulating phosphate metabolism, distal femur and calcaneus being the next most common sites. FD is a nonhereditary disease; a possible etiological cause may be represented by the mutation, during embryonic development, in somatic cells of the GNAS1 gene on chromosome 20q13.2–13.3. The incidence rates up to the 20s are explained by the fact that mutant cells decrease with age [94]. Mutations were found within coding region 8 of the gene when polymerase chain reaction analysis was used to amplify the patients’ genomic DNA [95].

#### 4.1.3. Clinical Features

Fibrous dysplasia most commonly presents among teenagers or young adults around their 20s. Lesions may occur in any bone, but its most common regions are the proximal femur, tibia, ribs, and skull. Fibrous dysplasia usually presents without any symptoms. However, in some cases, swelling may result in pain. When the deformity is severe, it leads to repeated pathologic fracture. One of the most common deformities caused by fibrous dysplasia is the “shepherd’s crook” varus deformity of the proximal femur that presents with pain and eventually limping.

#### 4.1.4. Microscopic Features

Histologically, fibrous dysplasia presents with a low to moderate cellular fibrous stroma surrounding irregularly shaped bone trabeculae without osteoblastic rimming.

#### 4.1.5. Imaging

In X-rays, fibrous dysplasia appears as a radiolucent, ‘‘ground-glass’’ expansive lesion that may present a bowing, without an evident trabecular pattern. There may be endosteal scalloping of the inner cortex, without periosteal reaction. Shepherd’s crook deformity consists of abnormal curvature of the femoral neck and proximal shaft (coxa vara) that may result in pathologic fracture. Periosteal reaction is usually absent unless there is a pathologic fracture [96]. 

#### 4.1.6. Differential Diagnosis 

Lesions that may resemble fibrous dysplasia include simple bone cysts, non-ossifying fibromas, osteofibrous dysplasia, adamantinoma, low-grade intramedullary osteosarcoma, and Paget disease. Simple bone cysts in X-rays appear more radiolucent than fibrous dysplasia because they are less dense, and surrounded by a thinner amount of lamellar bone with skeletal development, simple cysts move away from the metaphysis. It is possible to aspirate fluid from the cyst, which appears straw-colored. Non-ossifying fibromas are usually intracortical and eccentrically located in the growing metaphysis of weight-bearing long bones. NOFs spontaneously regress with age and generally present without pain. Non-ossifying fibromas can be distinguished from fibrous dysplasia because of the intracortical location, smaller size, lack of intralesional ossification, and spontaneous regression after skeletal maturity. Osteofibrous dysplasia or ossifying fibroma is usually localized almost exclusively to the distal third of the tibia or fibula. It usually occurs among children younger than ten years of age. Even though they are very similar to fibrous dysplasia, one key radiographic difference is that osteofibrous dysplasia usually has an intracortical location instead of the more central distribution of fibrous dysplasia.

#### 4.1.7. Treatment and Prognosis 

Gold standard treatment consists of bisphosphonates. Surgical treatment is considered in case of possible pathological fracture or persistent pain, spinal cord compression injury, and vertebral collapse instability. Biopsy is not indicated in case of characteristic radiographical findings. Series radiographs are advised to be performed every six months to assess progression absence [97]. Because of the common association with endocrine conditions, in the case of polyostotic disease, the patient must be tested for endocrine and metabolic tests for common associations with endocrine disorders. Surgical treatment consists of curettage and bone grafting, osteotomy with internal fixation to correct or even prevent bony deformities in extended lesions. FD growth is associated with skeletal development; therefore, it ceases after growth progresses slowly, and malignancies rarely occur. It is not typical for a malignant transformation to happen, with a prevalence ranging from 0.4 to 6.7%. The risk for malignant transformation increases in patients with PFD. Malignant transformation causes unexplained pain. In X-rays, it is visible osteolysis and mineralization of the lesion.

### 4.2. Non-Ossifying Fibroma

#### 4.2.1. Introduction

NOF is one of the five most frequent bone lesions and the most common benign lesion of the skeletal system [98]. According to the histologic classification of bone, it is classified by the World Health Organization (WHO) as a tumor-like lesion [99]. Several synonyms have been used, including histiocytic fibrous defect, metaphyseal fibrous defect, fibrous cortical defect, fibrous xanthoma, and histiocytic xanthogranuloma [60]. When multiple, non-ossifying fibromas could be presented with other extra-skeletal anomalies (syndromic) or not (non-syndromic). 

#### 4.2.2. Clinical Features 

Non-ossifying fibroma affects children and adolescents with an estimated prevalence of 30–40% of all children, mostly females. They are most commonly located in long bones, mainly in the distal femoral and distal tibial metaphysis, although they can also be found in the fibula and upper extremities. NOFs are asymptomatic and are detected only incidentally on radiographs.

#### 4.2.3. Macroscopic and Microscopic Features

Macroscopically, NOFs appear as fleshy, fibrous, yellow, or tan-brown lesions with variable areas of hemorrhage; microscopically, NOFs are characterized by the presence of swirls of fibrous tissue, containing multinucleated giant cells, hemosiderin, and foamy histiocytes [100]. 

#### 4.2.4. Sites of Development

Considering NOF of the distal femoral metaphysis, there is a clear relationship between tendinous/fibrous structures’ pull-on bone and their origin, therefore Ritschl et al. support that NOF lesions arise from tendons insertion into the perichondrium of the epiphyseal plate [101,102] as a tug lesion. This theory explains why NOFs arise primarily at the origin of the medial or lateral gastrocnemius or the insertion of the adductor magnus.

#### 4.2.5. Imaging

On radiographs, NOFs appear as solitary, eccentric and lytic lesions in the metaphysis of long bones, often polycyclic in shape. Typically, the lesion is lucent, and margins can appear densely sclerotic or indistinct, as well as the cortex, which may be thinned or, in some cases, thickened. When multiple, the most significant length of the lesion is oriented on the long axis of the bone [101]. NOF can spread, causing a thinning of the diaphyseal cortical bone with interspersed sclerosis and anterior or anterolateral bowing (Figure 4). 

#### 4.2.6. Differential Diagnosis 

The main entity often included in the histologic differential diagnosis of NOF is the focal cortical dysplasia (FCD): they refer to the same histopathological process, but NOF can be distinguished by its larger size >3 cm. The FCD is a small, lytic, intracortical lesion, typically eccentrically located and sharply outlined and represents an osteolytic defect with a thin shell of reactive bone. NOF is always eccentric and ovoid and often results in thinning and expansion of the overlying cortex. There is usually no periosteal reaction and no overt violation of the cortex. Simultaneously, NOFs can be radiographically distinguished by desmoids, which typically develop within the cortex; instead, NOFs are enlarged cortical defects that have expanded into the medullary cavity. Cortical desmoids are proliferative cortical irregularities. Histologically, desmoids are identical to NOF, so they differ only on radiographical presentation. 

#### 4.2.7. Treatment 

Asymptomatic lesions do not need any treatment. If an NOF involves more than 50% of the bone width on two simple X-ray views or more than 33 mm of length can result in fracture, so it needs a surgical approach. The treatment consists of local intralesional curettage or resection with bone graft filling [100].

## 5. Cystic Tumors

### 5.1. Unicameral Bone Cyst

#### 5.1.1. Introduction

Unicameral bone cysts, also known as solitary bone cysts, are lesions filled with fluid and lined with fibrous tissue. Among nonmalignant lesions, UBC is one of the most frequent, with a predominance of male over female of 2 to 1. UBC lesions are found in 80% of cases in teenagers or patients in their 20s [103]. UBCs development is thought to be associated with venous circulation disorders, specifically, blockage in the venous flow causes bone resorption because of increased pressure and an association with increased inflammatory protein levels in the intracystic fluid [104].

#### 5.1.2. Sites of Development

Unicameral bone cysts mainly affect the metaphyseal region of long bones, such as the humerus and femur.

#### 5.1.3. Clinical Features

Clinical behavior of UBCs is more aggressive in the first decade of life. In 80% of cases, UBCs are asymptomatic; otherwise, their presentation may vary from mild pain, local tenderness, and swelling to a pathologic fracture. Due to their mild or absent symptoms, they are typically discovered incidentally. 

#### 5.1.4. Micro-Macroscopic Features

Histologically, these lesions appear with a membrane with multiple ridges lining the inner surface of the cyst with a cementum-like material. The content is a serous or serous-bloody fluid, sometimes associated with hemosiderin, giant cells, or reactive new bone [105].

#### 5.1.5. Imaging

X-ray is the gold standard for diagnosis. The cyst arises from the medullary cavity, in the methaphyses of long bones, developing toward the diaphysis with the long axis parallel to the length of the bone. The lesion resembles a “rising bubble”, with this being a pathognomonic sign of UBCs (Figure 5).

#### 5.1.6. Treatment and Prognosis

Treatment consists of direct single or multiple injections of corticosteroids into the cyst itself. Bone grafting and curettage are two more invasive options reserved for larger bone cysts and those that compromise the integrity of the bone. The most appropriate treatments are chosen according to the patient’s age, size of the cyst, and whole bone strength [106]. About 15% of simple bone cysts during childhood heal without treatment, but most persist or enlarge [107].

### 5.2. Aneurismatic Bone Cyst

#### 5.2.1. Introduction

An aneurismatic bone cyst is a rare, benign blood-filled cyst neoplasm composed of vascular channels that tend to expand and, by doing so, can damage bone tissue. Usually solitary, it may arise spontaneously or accompany other benign tumors. It usually develops in children and young adults, mainly girls [108].

#### 5.2.2. Sites of Development

Mostly, ABCs are found in the posterior spine bones, proximal femur, tibia, and humerus, but they can arise from any bone. When affecting long bones, it develops preferably from the metaphyseal district [109].

#### 5.2.3. Micro-Macroscopic Features

According to their morphology, ABCs can be classified with the Capanna system [110] (Table 2):

Most of these lesions (70%), when primary, are associated with recurrent chromosomal translocations causing gene fusions with ubiquitin-specific peptidase 6 (USP6). The other 30% arise with other benign bone tumors [111].

#### 5.2.4. Clinical Features

Usually asymptomatic, ABCs can cause moderate pain, tenderness, and swelling. Their rapid growth can mimic a malignant tumor, that must be excluded. Other symptoms depend on the site of presentation: if it arises from the spine, it can cause neurologic deficits, torticollis, or scoliosis. Furthermore, it can cause pathologic fractures or growth disorders when arising close to growth plates [112].

#### 5.2.5. Imaging

Plain X-rays show lytic lesions with a sclerotic rim that sharply defines them. It can be the cause of pathologic fractures or periosteal reactions. A typical finding is the “soap bubble” appearance due to the presence of the remaining trabeculae. It is usually contained in the cortical bone. CT is valid for further study, especially in delicate districts like the pelvis and spine. MRI can show multiple blood-filled cavities. With contrast, septations inside the lesion can become more evident. Biopsy is necessary in any case for confirmation since its similar malignant behavior [113].

#### 5.2.6. Differential Diagnosis

It has to be done with unicameral bone cyst, giant cell tumor, telangiectatic osteosarcoma (malignant, shows tissue enhancement at MRI exam), osteoblastoma, and chondroblastoma.

#### 5.2.7. Treatment

The preferred treatment is surgical with resection, curettage, or bone grafting. Selective arterial embolization represents a valid treatment for central locations, difficult to access with other means. Other options are chemical cauterization, cryotherapy, and sclerotherapy with polidocanol and radionuclide ablation. To avoid too much blood loss during surgery, it is often performed preoperative embolization. Nonsurgical treatment with denosumab may be considered in cases when surgical excision is not a valuable option. Low dose of radiation can cause ossification of the lesion, albeit it is not used due to the risk of malignant transformation. Injections with different solutions like calcitonin and steroid under CT guidance, zein (corn protein), absolute alcohol, have been proven effective in some studies. This tumor, albeit benign, can be locally aggressive and destructive for its tendency to grow, so excision treatment represents a frequently chosen option. Cases of recurrence have been reported in the literature (10–20% after curettage) [114,115,116,117].

## 6. Conclusions

Benign bone tumors represent a frequent finding in everyday practice, especially in young patients. Due to their numerous common features, defined by clinical presentation, symptoms, and radiographic findings, it becomes mandatory for the clinician to know how to recognize them and act accordingly, avoiding misdiagnosis that can completely change patients’ therapeutic path. Even if many questions have been answered regarding these lesions, many are still open. We hope that further investigations will clarify the unknown sides of BBT.

## Figures and Tables

**Figure 1 jcm-11-00699-f001:**
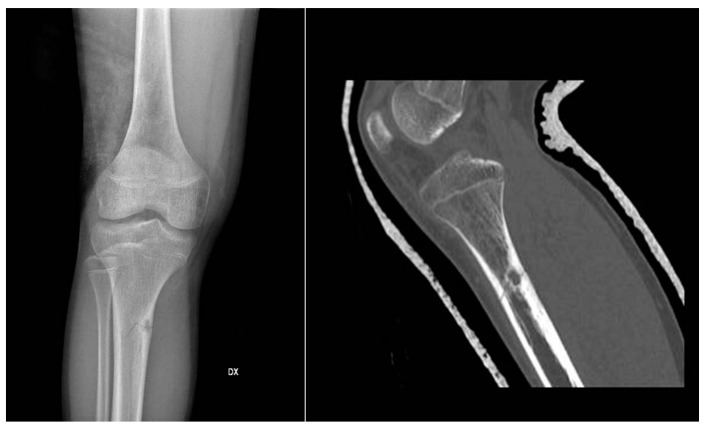
X-rays and CT scan of a tibial fracture that occurred in the development site of an Osteoid osteoma.

**Figure 2 jcm-11-00699-f002:**
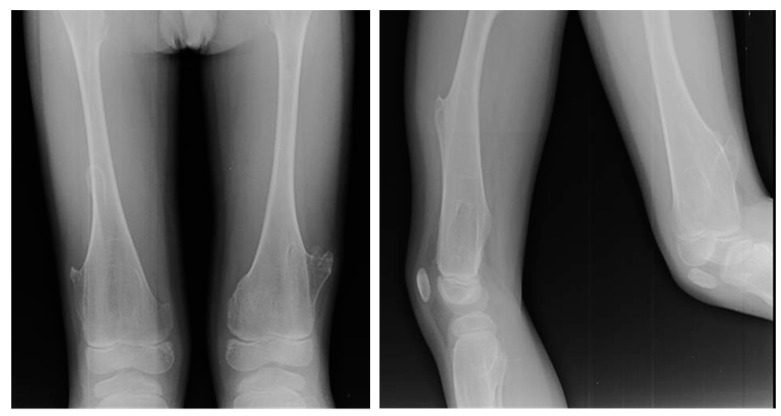
X-rays of multiple osteochondromas.

**Figure 3 jcm-11-00699-f003:**
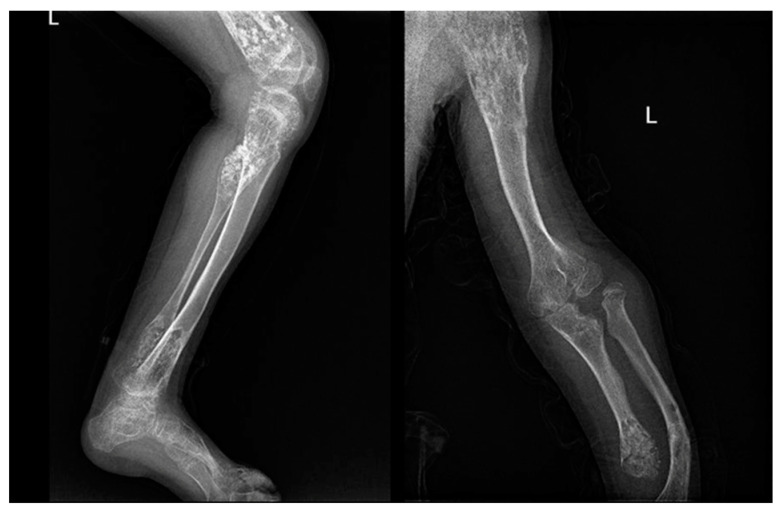
X-rays of a 10-year-old patient affected by Ollier syndrome.

**Figure 4 jcm-11-00699-f004:**
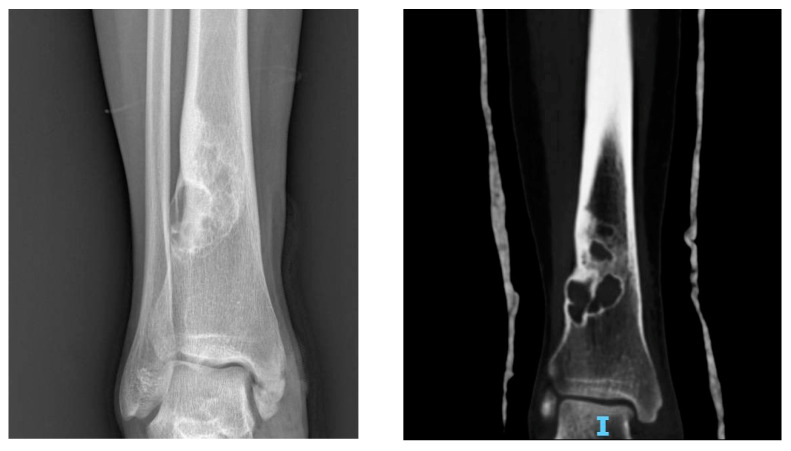
X-rays and CT scan of an NOF, incidentally discovered in a female patient in her 20s.

**Figure 5 jcm-11-00699-f005:**
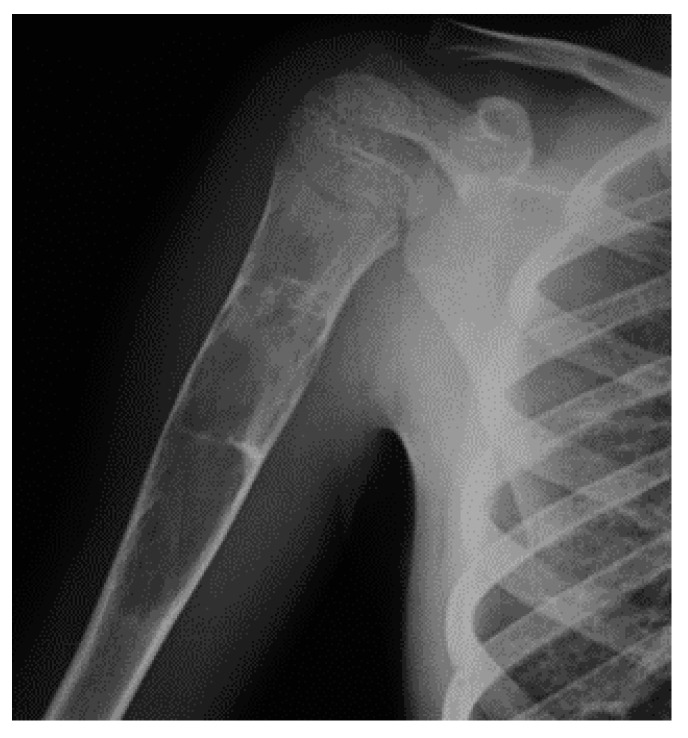
Unicameral bone cyst in a child’s humerus.

**Table 1 jcm-11-00699-t001:** Campanacci Grading.

Grade	Radiological Finding
I	Lesions confined in the bone cortex, which is intact (cystic lesions)
II	Lesions confined in the bone cortex, which appears thinned but not perforated
III	Lesions extended over the bone, that invade the cortex and expand to the soft tissues

**Table 2 jcm-11-00699-t002:** Capanna Classification System.

Type 1	Central well contained metaphyseal lesion
Type 2	Lesion involving an entire segment of bone with cortical expansion and thinning
Type 3	An eccentric metaphyseal lesion with no or minimal cortical expansion
Type 4	Subperiosteal reaction with no or minimal cortical erosion
Type 5	Periosteum displaced with the erosion of cortex and extension into cancellous bone

## Data Availability

No new data were created or analyzed in this study. Data sharing is not applicable to this article.

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
