# Peer review of "Benign Bone Tumors: An Overview of What We Know Today"

_jcm, 2022, doi:10.3390/jcm11030699_

Round 1
Reviewer 1 Report
the paper summarizes the main features of benign bone lesions.
I have suggested some clarifications and insights
in the attached document

Author Response
Q1. Figures: there are few figures, for osteoid osteoma only a tibial fracture where the nidus is harly seen on plain x-ray. I suggest to insert more figures, and of typical appearance of the tumor.
A1. We added better images both on plain x-rays and CT scan for osteoid osteoma as you suggested. Moreover, we included more images for the other lesions too.
Q2. Imaging of osteoblastoma: “Its appearance is similar to the osteoid osteoma neoplasm, with a round radiolucent 139nidus with well-defined margins surrounded by reactive sclerosis.”. actually the shape of osteoblastoma is usually not as regular as for osteoid osteoma, in particular not always round but can be somehow lobulated. Moreover the lesion contains fine radiodensities (new none formation from the neoplastic cells) which are more irregular and abundant if compared to osteoid osteoma (where there is a central, isolated core of radiodensity).
A2. Further pieces of information were added about the diverse imaging presentations of Osteoblastioma.
Q3. Treatment of osteoblastoma: “The most frequent choice is resection en bloc”. Actually being osteoblastoma a benign tumor the most used treatment is curettage. Resection is necessary for more aggressive cases, which don’t represent the majority of the cases.
A3. We made the requested correction.
Q4. Differential diagnosis for GCT: I should include iperparathiroidism in the differential diagnoses of GCT, being the bone lesions of this disease histologically very similar or identical to the true GCT.
A4. The differential diagnosis with brown tumors associated with hyperparathyroidism has been pointed out as you wrote.
Q5. “Metastases affect 2-3% of GCTB and develop most frequently from the lungs”. Distant metastases from GCT arise almost only in the lungs. The sentence “most frequently” suggests that metastases from GCT can also develop in other organs.
A5. The correction has been made.
Q6. Osteochondroma represents the most common bone tumor and is a developmental lesion rather than a true neoplasm. I would point out this aspect.
A6. We clarify this aspect. However more recent studies suggest that osteochondromas may truly be neoplasms as genetic mutations have appeared in both MHE and solitary forms [Pacifici M. The pathogenic roles of heparan sulfate deficiency in hereditary multiple exostoses. Matrix Biol. 2018 Oct;71-72:28-39.]
Q7. Is not clear what you mean with “…involves bone cortex and bone marrow..”. It should be more correct to write that the cortex of osteochondroma is in continuity with the cortex of the originating bone.
A7. As you suggest, we clarified the development of osteochondroma.
Q8. The radiographic appearance of osteochondroma is typical, so a standard radiograph is sufficient for the diagnosis. I would specify that further investigations are necessary in particular cases (in symptomatic patients): bursitis, impingement with vascular-nervous structures, growth in the adult with suspicion of malignant transformation…
A8. There was a typing mistake
Q9. Sublingual instead of subungual
A9. As you suggest, we add the request
Q10. Since osteochondroma derives from growth plate cells, it would be important to specify that recurrence after excision is possible only during body growth, if the entire cartilage cap is not removed.
A10. As you suggest, we added the pieces of information.
Q11. The risk of transformation in chondrosarcoma is still low even in multiple hereditary exostoses. If you found a so high incidence can you cite the bibliography on that data?
A11. There was a typing mistake.
Q12. Actually enchondroma is formed, composed by hyaline cartilage, more 2 then it involves cartilage.
A12. As you suggest, we clarified the nature and the development of enchondroma
Q13. The size of enchondroma vary from < then 1 to very large lesions (in rare cases the lesion may occupy the whole diaphysis). The sentence “with a diameter <5 cm” implies that they are always smaller than 5 cm.
A13. We removed the data about diameter
Q14. Pain may also suggest malignant transformation of a benign enchondroma
A14. As you suggest, we added the information.
Q15. Calcifications inside an enchondroma form during the adulthood (along the years), I would specify this
A15. We specified the period in which calcification appears
Q16. The classification proposed by Takigawa is for enchondromas of the hand, not for all enchondromas. Specify this, or remove the citation of this classification. Moreover, is a paper written in 1971, reporting results of cases treated from 1956. Perilesional edema is more likely expression of malignancy, anyway this concept is not reported in the reference 76.
A16. We removed Takigawa classification and its citation.
Q17. Differential diagnosis between enchondroma, and chondrosarcoma is a challenge, expecially if chondrosarcoma is atypical cartilaginous tumor or low grade. I recommend to be more in-depth on this topic (clinic and radiological differences).
A17. Further informations were added about differential diagnosis
Q18. Curettage and cement makes it easier to identify eventual local recurrence (in some cases the true diagnosis -benign enchondroma/atypical cartilaginous tumor/grade 1 chondrosarcoma- is made after curettage)
A18. Thank you for your clarification.
Q19. FD is not a true tumor but a dysplastic lesion. I would change the word “tumor” with “lesion”
A19. The word tumor has been changed with the word lesion
Q20.“MFD generally grows along with skeletal development, eventually it stops, resulting in new pathological fractures or deformities”. From this sentence it seems that monostotic fibrous dysplasia at the end of growth still causes a pathological fracture. I recommend rewriting the sentence more clearly (when it causes path fracture?)
A20. As you suggested, the concept has been clarified
Q21.specify Mazabraud syndrome (FD + multiple soft tissue mixomas)
A21. Mazabraud syndrome has been described
Q22. In case of deformities, corrective osteotomy and fixation may be indicated
A22. Further information about treatment has been added
Q23. A very useful and diffused treatment for ABC is selective arterial embolization (for central locations). I would cite it. Embolization is usually used as a treatment rather then a preoperative procedure in ABC.
A23. The description of the treatment has been added.
Reviewer 2 Report
- English needs improvement, try revising the manuscript with help from a native English speaker
- In osteochondromas, the role of MRI is also the evaluate the thickness of the cartilage cap. This should be added.
- In enchondromas, the tumor is radiolucent on radiographs and CT with internal cartilage matrix (rings and arcs calcifications), this should be added.
- There are 2 Figures 2 (Ollier syndrome and NOF). Please revise.
- It would be nice to have more images in the article including MRI images. An MSK radiologist should be able to provide these.
Author Response
Q1. English needs improvement, try revising the manuscript with help from a native English speaker
A1. Thank you for your suggestion, the manuscript was proofread by an English speaker.
Q2. In osteochondromas, the role of MRI is also the evaluate the thickness of the cartilage cap. This should be added.
A2. We added the pieces of information you requested.
Q3. In enchondromas, the tumor is radiolucent on radiographs and CT with internal cartilage matrix (rings and arcs calcifications), this should be added.
A3. We provided to add this information.
Q4. There are 2 Figures 2 (Ollier syndrome and NOF). Please revise.
A4. We revised this mistake.
Q5. It would be nice to have more images in the article including MRI images. An MSK radiologist should be able to provide these.
A5. We provided by adding more images.
Round 2
Reviewer 2 Report
thank you for the revisions